# Targeted Sequencing of Mitochondrial Genes Reveals Signatures of Molecular Adaptation in a Nearly Panmictic Small Pelagic Fish Species

**DOI:** 10.3390/genes12010091

**Published:** 2021-01-13

**Authors:** Miguel Baltazar-Soares, André Ricardo de Araújo Lima, Gonçalo Silva

**Affiliations:** MARE-Marine and Environmental Sciences Centre, ISPA-Instituto Universitário, Rua Jardim do Tabaco 34, 1149-041 Lisboa, Portugal; alima@ispa.pt (A.R.d.A.L.); gsilva@ispa.pt (G.S.)

**Keywords:** small pelagic fishes, OXPHOS complex, adaptive potential, climate change

## Abstract

Ongoing climatic changes, with predictable impacts on marine environmental conditions, are expected to trigger organismal responses. Recent evidence shows that, in some marine species, variation in mitochondrial genes involved in the aerobic conversion of oxygen into ATP at the cellular level correlate with gradients of sea surface temperature and gradients of dissolved oxygen. Here, we investigated the adaptive potential of the European sardine *Sardina pilchardus* populations offshore the Iberian Peninsula. We performed a seascape genetics approach that consisted of the high throughput sequencing of mitochondria’s *ATP6*, *COI*, *CYTB* and *ND5* and five microsatellite loci on 96 individuals coupled with environmental information on sea surface temperature and dissolved oxygen across five sampling locations. Results show that, despite sardines forming a nearly panmictic population around Iberian Peninsula, haplotype frequency distribution can be explained by gradients of minimum sea surface temperature and dissolved oxygen. We further identified that the frequencies of the most common *CYTB* and *ATP6* haplotypes negatively correlate with minimum sea surface temperature across the sampled area, suggestive of a signature of selection. With signatures of selection superimposed on highly connected populations, sardines may be able to follow environmental optima and shift their distribution northwards as a response to the increasing sea surface temperatures.

## 1. Introduction

Ongoing climatic change translates into a series of environmental fluctuations with the potential to transform the current distribution of global biodiversity. As biological responses fall in the spectrum of “adapt, move or go extinct”, the range shifts, biological invasions and re-shaping of entire ecosystems, are thus expected to increase in frequency [1]. The marine environment is particularly susceptible to climatic shifts [1], and phenomena such as rising sea surface temperatures, increases in CO_2_ concentrations (ocean acidification) and depletion of O_2_ in deep oceanic waters, translate into tropicalization of fish assemblages in temperate zones [2,3] and hypoxia in coastal regions and estuaries [4,5]. Investigating the adaptive potential of species to cope with environmental shifts relies on the identification of phenotypic or molecular changes, expected to emerge as evolutionary responses [6]. Advances in sequencing technologies have greatly improved research on the molecular variation across non-model organisms [7]. Coupled with the extensively available environmental data that exists in marine regions, seascape genetic approaches are frequently being used to identify signatures of evolutionary processes [8,9].

On top of the environmental pressures imposed by climate change, marine species of commercial relevance are also subject to overexploitation. Unsustainable harvesting compromises the viability of stocks in multiple ways. For example, underestimating stock sizes leads to a logical reduction in the critical mass of spawners necessary to maintain population dynamics [10]; capturing larger individuals imposes an early age at maturation, reducing the overall quality and quantity of brood stock [11,12]; disproportionally harvesting of unequal stocks may eliminate unique genetic diversity, thus reducing the species with adaptive potential, and therefore limiting the breath of possible responses to environmental pressures [13]. Coastal species with commercial value are heavily exposed to a synergistic effect of environmental and anthropogenic pressures. In this context, stocks of small pelagic fishes such as sardines, anchovies or herrings are among those that are particularly vulnerable [14]. “Small pelagic fishes” defines a group of ray-finned and fusiform short size fishes that tend to form large shoals that inhabit the water column near coastal locations. They mostly aggregate at the edge of continental shelves in order to benefit from the oxygen and nutrient-rich waters that emerge during upwelling events [15,16]. Due to their abundance, biomass, and trophic level occupied, those fish are an important component of an ecosystem that expands beyond the coastal limitations of their distribution [17,18,19]. 

The European sardine *Sardina pilchardus* is a small epipelagic fish inhabiting the north eastern Atlantic coast, the Mediterranean, and the Black Sea. It is a key component of the Mediterranean blue economy, supported by centuries-old tradition of consumption, and more recently, production of canned products [20]. Both the Mediterranean and Eastern Atlantic stocks along a costal line from Senegal to Galicia, in Spain, are in a particularly fragile state [21,22,23]. Consequently, sardine fisheries have a recent, yet long record of fishing restrictions, quota limitations, and premature closures of fishing seasons. 

In this work, we aim to investigate the adaptive potential of sardine populations surrounding the temperate zone of the Iberia Peninsula to fluctuations on sea surface temperature and dissolved oxygen. We focused on genetic variation at the mitochondrial level, as the mitochondrial genome encodes for genes involved on the oxidative phosphorylation (OXPHOS) pathway [24,25]. The central role of mitochondria in organismal metabolism questions longstanding assumptions of the organelle’s evolutionary neutrality [26]. It is thus not surprising that signatures of selection have been identified among the mitochondrial-encoded OXPHOS genes in several fish species, including Atlantic and Pacific salmons, anchovies, or sardines in the Indian Ocean, and associated with thermal gradients or dissolved oxygen concentration [26,27,28]. We followed a target-gene approach to screen the molecular diversity of sardine’s mitochondrial ATP6, CYTB, ND5 and COI, which code for complexes I,III–V of the OXPHOS pathway, and specifically sequenced key functional regions where signatures of selection have been detected in other marine fishes [26,27]. We also sampled a broad geographic area, enveloping the coastal waters of all the Iberian Peninsula, and consequently a gradient of coastal environments and putative barriers to dispersal. To explore the hypothesis that the frequency of mitochondrial variants among mitochondrial-encoded OXPHOS genes is related to the spatial variation of temperature and dissolved oxygen concentration, we collated environmental information from sampling locations. Finally, we complemented the survey of the species’ genetic diversity with five microsatellite markers, which allowed us to inspect the connectivity among sampled sites. 

## 2. Methods

### 2.1. Sample Collection and DNA Extraction 

A total of 96 individuals from 5 locations, Bay of Biscay (Spain), Gulf of Lion (France), Sesimbra (Portugal), Olhão (Portugal) and Tarragona (Spain) were collected, spanning the Atlantic-Mediterranean region surrounding the Iberian Peninsula (Figure 1). DNA was extracted with EZ-DNA Genomic DNA Isolation kit© (Zymo Research, California, CA, USA) following manufacturer instructions. Quantity and quality of extracted DNA were first assessed with Nanodrop© and later with Qubit© fluorometer.

### 2.2. Mitochondrial DNA: Defining Target-Regions for Amplicon Sequencing 

To explore whether the mitochondrial regions identified as putative under selection on other small pelagic fishes would exhibit similar signatures on sardines, 4 mitochondrial genes, each encoding different subunits involved in specific complexes of the mitochondrial respiratory chain were used: ND5 (NADH: ubiquinone oxireductase core subunit 5) for respiratory complex I; CYTB (cytochrome b) for respiratory complex III; COI (cytochrome c oxidase I) for respiratory complex IV; ATP6 (ATP synthase membrane subunit 6) for respiratory complex V. To facilitate the identification which within-gene regions should target sardines, we collected sequences from NCBI of other small pelagic species on which signatures of selection have been reported on our target genes. Those were then aligned with BioEdit [29] against the sardine mitogenome (reference NCBI: NC_009592.1) to verify where matching occurs. The conservation of chosen regions was further checked against other fish species. Primers were designed on the sardine genome to flank putative regions of interest. Reference numbers of utilized sequences are available on Appendix A.

#### 2.2.1. Sequencing, Filtering and Variant Calling at Targeted Mitochondrial Regions

Extracted DNA was sent to CIBIO (Centro de Investigação em Biodiversidade e Recursos Genéticos) in Vairão, Portugal, to primer optimization, library preparation and amplicon sequencing. The primer-pairs specifically designed for this study are available on Appendix A. All obtained sequences are available in Appendix A. Detailed protocols on library preparation and amplicon sequencing are available in Section A.1.

Demultiplexing, barcode removal and QC control were performed in Illumina’s Basespace online platform. Only reads with Phred quality score (Q) > 30 were utilized in this study. Demultiplexed reads were aligned against a reference, i.e., sardine’s mitogenome (reference NCBI: NC_009592.1) with *bwa-mem* algorithm [30] implemented in bwa [31]. Only reads that were properly paired were kept, mapping qualities above 10 and coverage above 2 were kept for variant calling. Variant calling was performed with Stacks v 1.48 [32] through imported .bam files. Specifically, we ran ref_map.pl pipeline with the sardine mitogenome as reference and default parameters. 

#### 2.2.2. Diversity and Distribution of Genetic Variation at Targeted Mitochondrial Regions

Haplotype diversity (Hd), number of haplotypes (nH), nucleotide diversity (π) and segregation sites (S) were calculated both independently for each gene in DNAsp [33]. In total, 4 datasets, corresponding to each sequenced mitochondrial region were built. To assess the connectivity among sampled locations, the population structure in Arlequin for all datasets was estimated via F_ST_ calculation (10,000 permutations). The level of significance was set to α = 0.05 [34]. Isolation by distance was inferred by using log-transformed distance-by-sea among sampling locations and F_ST_ pairwise values in the R package ecodist, with the level of significance set to α = 0.05 [35]. The number of shared haplotypes among locations was calculated to inspect whether geographic distance could play a role in the distribution of haplotype diversity. Finally, median-joining networks were calculated and drawn to better understand the genealogy among identified haplotypes in POPART [36].

#### 2.2.3. Relationship between Haplotype Composition and Environmental Variables at Each Location 

All computations were performed in the R environment [37]. A Redundancy Analysis (RDA) was performed to investigate whether haplotype distribution could be explained by environmental conditions found at each collection site. Environmental parameters (annual monthly means) were extracted from the Bio-Oracle database [38] with scripts customized to collect data on the maximum, minimum, mean and the range of values of sea surface temperature (SST) and dissolved oxygen (DO) given the approximate GPS positions of exact sampling locations. Thus, a matrix of haplotype frequencies was utilized as a response variable in an RDA [39], while a matrix of values related to surface temperature and dissolved oxygen were utilized as putative explanatory variables. Haplotypes unique to single locations were excluded from the dataset for this analysis. 

The overall statistical significance of the RDA model was assessed using an ANOVA-like permutation function, anova.cca, implemented in the in the R package *vegan* [40]. Significance of constrained axis and effects and significance of each variable were also assessed with anova.cca, defining the “axis” and “margin” to avoid the sequential test of terms [41]. Lastly, a putative association of specific haplotype frequencies and environmental variation was inferred by considering, as significant, those relationships that diverged 2.5 standard deviations from the mean distribution on the RDA plot (i.e., two-tailed *p*-value = 0.0125). 

#### 2.2.4. Inferring Historical Events of Selection across Phylogenies with dN/dS Ratios-Based Tests

As we were interested in screening for signatures of selection on each specific respiratory complex, a phylogenetic inference of events of selection was performed independently for each gene-specific dataset. As such, we utilized methods included in Hyphy via the Datamonkey web interface, which are based on the inference of nonsynonymous and synonymous substitution rates across phylogenies [42]. A single-likelihood ancestor counting (SLAC) [43], a fixed effect likelihood (FEL) [43] and fast unconstrained Bayesian approximation (FUBAR) [44] algorithms were applied. While sharing the same theoretical basis, i.e., selective pressure for each site is constant along the entire phylogeny; algorithms vary on the methodology to infer dN and dS substitution rates. Specifically, SLAC uses the maximum-likelihood and counting approach, FEL uses a maximum-likelihood approach and FUBAR relies on a Bayesian framework. Significance was assessed accordingly to the default value of each specific test, namely, with the level of significance set to α = 0.1 (rounded-up to the decimal) in likelihood models SLAC and FEL, and with a posterior probabilities threshold of >0.9 (rounded-up to the decimal), for (α > β) in FUBAR (default). 

### 2.3. Microsatellite Amplification and Diversity Estimates

Nuclear genetic variation was investigated with five microsatellites previously described in other sardine studies–SAB07, SAR09, SR15, SAR112, SAR218, and SP17 [45,46]. Amplification and genotyping protocols can be found in Section A.1. Nei’s unbiased heterozygosity (He), observed heterozygosity (Ho), Wright’s inbreeding coefficient (FIS) and Garza-Williams’ index were computed in Arlequin v 3.5.2.2 [34]. Rarefied allelic richness (Ar) and private rarefied allelic richness (pAr) were estimated for both each location and averaged over loci in HP-Rare 1.1 [47]. The presence of null alleles was inspected with Micro-Checker [48].

#### 2.3.1. Estimates of Genetic Differentiation and Structure among Sampled Locations

We first assessed the likelihood of each locus to be in the Hardy-Weinberg equilibrium (HWE), as consistent deviations could compromise assessment of neutral differentiation. HWE was computed per loci and population in Arlequin v 3.5.2.2. The possibility of population structure was then investigated through multiple approaches. First, we utilized a Discriminant Analysis of Principal Components (DAPC) to explore the number of clusters (*K*) that most likely explains the observed distribution of nuclear diversity implemented in the R package adegenet [49]. This was achieved through the application of the function find.clusters with max.n .clusters = 5, identification of most likely *K* through the inflection point of the Bayesian Inference Criteria (BIC) curve and subsequent calculation of discriminant components. Lastly, we plotted the 95% confidence interval ellipses of the identified *K* groups overlaid on the original distribution. For that, we utilized the two major linear discriminants (LD1 and LD2) extracted from the discriminant analysis as graphical coordinates. We further estimated the pairwise F_ST_ among locations, derived from allelic frequencies in Arlequin v3.5.2.2 (10,000 bootstrap). Isolation by distance was also inspected and analyses were performed similarly to those described in the previous Section 2.2.2. 

#### 2.3.2. Inferences on Deviations from Evolutionary Neutrality 

Additional locus-specific analyses were performed because one of the utilized markers (SAR112) was reported to be the subject of clinal selection along the Mediterranean [46]. In this context, first, the frequency of HWE deviation across locations for each locus was explored. Then, the locus-specific allele frequency distribution among populations was analyzed by comparing the average F_ST_ obtained from respective pairwise comparisons. Frequencies were compared with ANOVA and Tukey’s post-hoc tests being applied to investigate the significance of pairwise relationships. 

## 3. Results

### 3.1. Mitochondrial DNA Sequencing and Variant Calling

The 96 individuals sequenced for the four targeted genes produced a total of 912,609 usable reads. On average, 2223.84 (SE ± 128.96) paired-end reads per individual were utilized for the ATP6, 2599.17 (SE ± 115.99) for the COI, 2693.29 (SE ± 140.73) for the CYTB and 1990.04 (SE ± 97.33) for the ND5. Average coverage per individual varied from 304.07 (SE ± 18.80) on ATP6 to 208.19 (SE ± 9.22) at CYTB. The position of variant sites identified with *Stacks* was inspected visually and utilized to construct haplotypes from the reference mitogenome. As genes were analyzed as fully independently, fragments were trimmed to remove uninformative invariant sites at the edges of the DNA string prior to population genomic analysis. Sizes of the fragments utilized varied across genes, specifically ATP6 (381 bp), COI (291 bp), CYTB (201 bp) and ND5 (330 bp). 

### 3.2. Estimates of Mitochondrial Genetic Diversity and Population Structure

Genetic diversity indices expectedly varied among genes. Here, CYTB exhibited the higher indices of diversity (µ_S_ = 3.6, SE_S_ = 0.219; µ_nHap_ = 4.6, SE_nHap_ = 0.219; µ_Hd_ = 0.717, SE_Hd_ = 0.02; µ_π_ = 6 × 10^−3^, and SE_π_ = 3 × 10^−4^), while the COI showed a high conservation rate (µ_S_ = 1.4, SE_S_ = 0.219; µ_nHap_ = 2.4, SE_nHap_ = 0.219; µ_Hd_ = 0.164, SE_Hd_ = 0.02; and µ_π_ = 5 × 10^−4^, SE_π_ = 8 × 10^−5^). Among the sampled locations, genetic diversity was only pronouncedly different for the ATP6, as Tarragona and the Gulf of Lions reported a single haplotype in contrast to the Bay of Biscay and Sesimbra, where we observed three different haplotypes (Table 1). Estimates of pairwise F_ST_ only revealed two significant differentiation values, specifically for ATP6 haplotype frequencies between Sesimbra and the Gulf of Lions (F_ST_ = 0.045, *p* = 0.042) and ND5 between Tarragona and Olhão (F_ST_ = 0.114, *p* = 0.033). Investigating the overall number of haplotypes shared among locations revealed that closer sampled locations shared more haplotypes, suggesting that geographic distance might play a role in connectivity, despite the absence of clearly structured populations (Figure 2) and non-significant Mantel tests across genes (Appendix A).

### 3.3. Relating Haplotype Frequencies and Environmental Variation

The low number of different haplotypes in relation to explanatory variables prevented us building a single, full-scale model to incorporate maximum, minimum and averages of both dissolved oxygen and sea surface temperature. However, to preserve the fundamental objective of exploring and disentangling the role of those variables in sardine’s distribution, we built four RDA models for each specific set of minima, maxima, mean and range. Only the model that incorporated minima was revealed to be statistically significant (ANOVA: F = 2.84, d.f. = 2, *p* = 0.05), and with an adjust R^2^ = 0.48 (Figure 3A) (Table 2). CAP1 was revealed to significantly explain 43% of observed variability (ANOVA: F = 3.343, *p* = 0.050) and the marginal effect of each environmental variable showed minimum sea surface temperature (sst_min) to be significant (ANOVA: F = 3.109, *p* = 0.033) (Table 3).

We further identified haplotype 1 of ATP6 and haplotype 3 of CYTB to present a significant negative correlation with (minimum) sea surface temperature (Figure 3b). Models with average and maxima dissolved temperature and sea surface temperature did not significantly explain the haplotype frequencies distribution (Table 2).

### 3.4. Inferences of Episodic Selection across the Phylogeny of Sampled Specimens

Investigating the occurrence of pervasive selection across the entire gene phylogenies revealed sites deviating from a neutral evolution background on all our target genes. SLAC detected six sites on CYTB, three of which revealed signatures of positive or directional selection and three others on negative or purifying selection. FEL and FUBAR, whose statistical power is more robust than SLACs, both identified episodes of pervasive selection on genes ATP6 and ND5. For ATP6, FEL detected site 53 (*p* < 0.1), while FUBAR detected sites 53, 49, 24, 36, and 52 (posterior probability of α > β, ≥0.9). For ND5, FEL detected site 58 (*p* = 0.05), while FUBAR detected sites 58, 64, and 90 (posterior probability of α > β, ≥0.9). The Bayesian approach further identified four sites on the COI, namely, 8, 44, 53, and 57 (posterior probability of α > β, ≥0.9) (Table 4). All these signatures relate to negative/purifying selection.

### 3.5. Microsatellite Amplification and Diversity Estimates

In general, we did not detect prominent variation regarding population specific diversity. Observed heterozygosity (Ho) values ranged between 0.65 (SD ± 0.27) in Sesimbra and 0.74 (SD ± 0.16) for the Gulf of Lion. Rarefied allelic richness over all loci varied between Sesimbra (12) and the Bay of Biscay (13.44). Private allelic richness, and indicator of population-specific alleles, was also similar from one location to another, being the extremes Tarragona (1.18) and the Bay of Biscay (2.24). Garza-Williams index, which estimates the effects of potential bottlenecks from observed heterozygosity, also indicated a striking similarity across populations (Table 5). Observed and expected heterozygosity (He) were revealed to significantly differ across sampled locations (ANOVA: F_1,4_ = 6.02, *p* = 0.01) (Appendix A).

### 3.6. Estimates of Microsatellite Genetic Differentiation and Structure among Sampled Locations

Inferences on the structure of sardine populations revealed a nearly panmictic scenario. Discriminant component analyses suggested a K = 2, though 95% CI ellipses largely overlap (Appendix A). Pairwise F_ST_ comparisons further reinforced such a scenario as all but one showed a significant result, for *p* = 0.02 (F_ST *Bay of Biscay vs Gulf Lion*_ = 0.05). While those locations are at the maximum geographic distance possible within our sampled sites, correlating geographic and genetic distances obtained no significant value. Lastly, STRUCTURE analyses also revealed panmixia (Appendix A).

Regarding locus-specific analyses, we first detected locus SAR218 to be constantly departing from Hardy-Weinberg equilibrium: it was the only locus whose *Ho* significantly deviated from *He* in all populations. This is probably due to the presence of null alleles at this locus, which was found to be consistent across all examined populations (Appendix A).

Exploring the structure and explaining each locus, we did not observe significant departures from what would be expected from panmixia: CIs of the 95% ellipses largely overlapped in every locus (Appendix A). Pairwise F_ST_ analyses suggested a very similar picture, with only one significant pairwise comparison between Sesimbra and Olhão for the locus SAR112 (F_ST_ = 0.013, *p* = 0.036). However, we detected averaged pairwise F_ST_ among comparisons and this varied significantly across loci (ANOVA: F_1,5_ = 21.385, *p* < 0.01) (Figure 1B). Post-hoc comparisons further revealed locus SAR218 to be the one originating in significantly higher average F_ST_ than all the other loci, Tukey HSD (Avg_*SAR218*_ = 0.023, SD_*SAR218*_ = 0.005). 

## 4. Discussion

Environmental alterations driven by the ongoing rapid climatic shifts impose intense selective pressures on populations of marine organisms, and particularly on those that are currently exploited. Understanding the response repertoire of those species is crucial to develop effective and sustainable exploitation plans to accommodate potential shifts in the distribution range and overall impacts on the viability of populations. By screening the variation on genes directly involved in metabolic performance, we aimed to infer the adaptive potential of sardine populations to shifts in temperature and dissolved oxygen. We observed a high degree of connectivity among populations and further revealed a putative molecular basis of adaptation to cold temperatures. Under the predicted increase in sea surface temperature, our results indicate that sardines might be able to follow environmental optima and extend their distribution northwards. 

### 4.1. Population Structure Reveals High Connectivity around Iberian Peninsula

Genetic diversity estimates across investigated markers suggested the Bay of Biscay harboring higher diversity among investigated populations. At the mitochondrial level, three out of four genes (CYTB being the exception) exhibited a higher number of haplotypes and/or nucleotide diversity, and microsatellite markers consistently showed a higher rarefied number of alleles and private alleles among sardines collected on the Bay of Biscay. The expected a positive correlation between genetic diversity and effective population size and may simply suggest that the Bay of Biscay assembles a larger number of reproducing individuals in comparison to all others [50]. The results are coherent with observations reporting the Atlantic sardine stocks holding higher abundances than those in the Mediterranean [51,52,53]. Alternatively, differences among observed diversity estimates may be associated with variable fishing efforts in relation to stock size, as overfished stocks are predicted to exhibit lower genetic diversity than those that are sustainably managed [54]. Population structure analyses revealed a scenario of near panmixia, punctuated by deviations likely associated to geographic distance among sampled sites. Specifically, not only nearby locations shared a higher number of haplotypes, but also the F_ST_ pairwise estimate associated with microsatellite data between the Bay of Biscay and the Gulf of Lion. As Mantel tests did not support a general pattern of isolation by distance, we consider having identified a signature of limited dispersal among sampled locations, possibly due to the natural barrier imposed by the Alboran Sea, as suggested by Ramon and Castro (1997) [55].

Generally, our observations are consensual within the existing studies dedicated to sardine population structure in the region, though the majority suggests a sardine panmictic stock [22,51,56], and others point towards a soft structure on a specific region of the species distribution [55]. While our study did not aim to exhaustively characterize the population structure of the species, we believe that it reinforces the need to undertake genome-wide approaches and temporal sampling to convincingly resolve the stock structure of the European sardine. 

### 4.2. Temperature-Dependent Distribution of Most Frequently Observed Haplotypes 

Exploring the role of specific environmental factors on the distribution of sardine’s mitochondrial diversity revealed that only minimum values of temperature and dissolved oxygen across sampled locations significantly explained the gradients of genetic variation. By showing that the distribution of the more frequently observed haplotypes of two of the targeted genes—ATP and CYTB—negatively correlated with minimum surface temperature, we can infer that the sardine stock in the screened geographical area is selected to tolerate colder sea surface temperatures. These results agree with previous works on mitochondrial DNA selection on other marine fishes, particularly on anchovies [27] and on Atlantic salmon [26], probably indicating a functional need to increase metabolic efficiency at lower temperatures [26]. North eastern Atlantic latitudes covered by our work are characterized by a strong seasonal upwelling, that creates oxygen- and nutrient-rich environments nearby coastal costal locations but originates from lower sea surface temperatures [57]. One of the anticipated effects of ongoing climate change on coastal upwelling systems is an increase in intensity of spring and summer upwelling events [57]. As the Iberian Peninsula is particularly sensitive to increases in upwelling intensity [58], we interpret the relationship identified in our work as a putative genomic signature of the species’ response. Interestingly, historical signatures of purifying selection identified amongst the phylogenies of targeted genes were associated with low frequency haplotypes. As all variants are exposed to selection in haploid systems, such observation suggests the evolutionary dynamics of the sardine genetic pool purged specific haplotypes from the population due potentially deleterious effects [59,60]. While synonymous mutations do not alter the sequencing of encoded protein, they are known to impact secondary structures affecting thermodynamics and stability, as observed on mRNA molecules [61], misfolding of proteins [62] or translational selection and codon use bias in both proto and metazoans [63,64]. In addition, and specific to small pelagic fishes, the often large effective population sizes would reduce the efficiency of genetic drift on removing rare variants, but see Cvijović et al. (2018) for alternative theoretical expectations on site-frequency distribution under puryfying selection [65]. Lastly, genetic signatures on mitochondrial genes of the OXPHOS complexes are also worth exploring at the light of cyto-nuclear co-evolution [66] or having their functional impact on organismal fitness effectively assessed given the existence of compensatory mechanisms to buffer the expression of deleterious alleles [67]. 

## 5. Conclusions and Future 

Under and intense fishing pressure, the capacity for the European sardines to positively respond to ongoing climatic shifts must be resolved before developing practices of sustainable exploitation. Here, we focused on the European sardine stocks around the Iberian Peninsula and provided additional evidence for the existence of an apparent near panmictic population with a high degree of connectivity. This connectivity might be important to maintain frequency distribution of key mitochondrial haplotypes across the species distribution range, ensuring not only the species adaptive potential, but also an overall capacity to migrate to follow environmental optima: eggs and larvae of European sardine have been reported as North as the Baltic Sea [68]. Strong northward shifts are nevertheless expected for temperate-cold water species such as European sardines, according to species distribution models under future climatic conditions [69]. 

Shifts in thermal conditions are particularly stressful for ectotherms. Additionally, there are certainly other mechanisms involved in a putative adaptive response to thermal variation. For instances, adaptive and non-adaptive phenotypic plasticity are known to play a decisive role in assisting organisms to cope with shifting temperatures [70,71]. Epigenetic variation is also another possible route towards adaptation, and evidence of such keeps on accumulating among both natural populations and under controlled experiments [72,73,74]. Though we interpreted the relationship between haplotype frequency and minimum sea surface temperature as a signature of selection, we cannot disregard the potential effect of small sample sizes or the fact that we did not screen the edges of specie’s distribution. Still, with the seascape genetics approach we undertook, we are confident that our study provides a robust glimpse of sardines’ adaptive potential and is an important contribution to the growing body of evidence supporting adaptation among populations of small pelagic fishes. 

## Figures and Tables

**Figure 1 genes-12-00091-f001:**
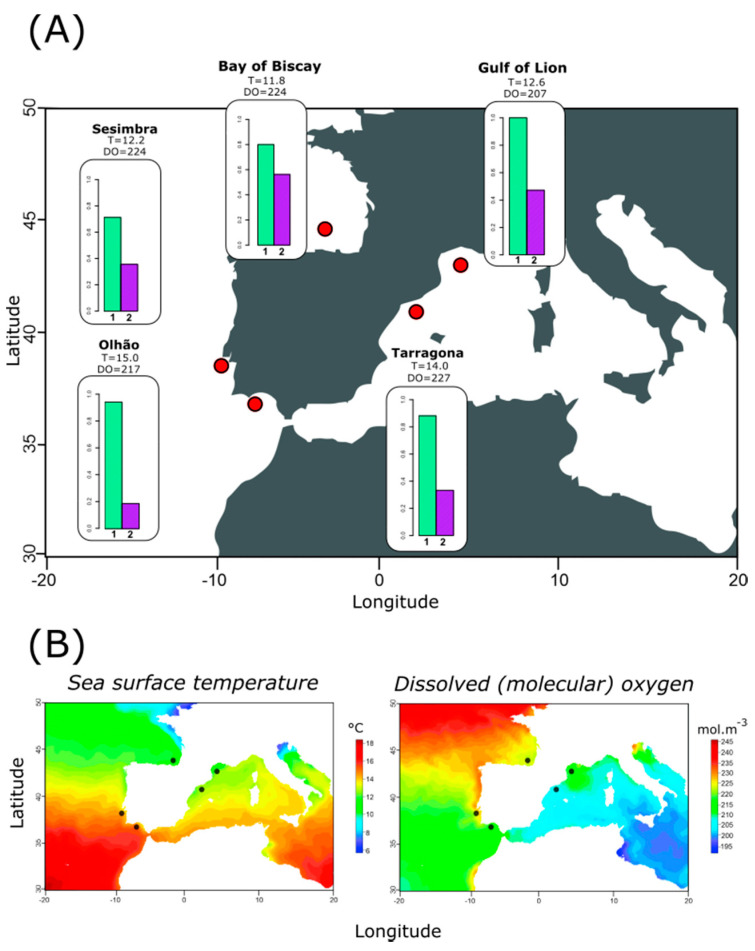
Sampling locations and associated environmental variation. (**A**) Sampling locations and plots of the frequencies (*y*-axis = frequency of the most common haplotype among ATP6, ATP6-Hap1 (**1**) and CYTB, CYTB-Hap3 (**2**). The minimum sea surface temperature (T °C) and minimum dissolved oxygen (mol·m^−3^) obtained from each location is also shown. (**B**) Heatmap of minimum sea surface temperature and minimum dissolved oxygen distribution in the marine area surrounding the Iberian Peninsula. Black dots represent the sampling locations.

**Figure 2 genes-12-00091-f002:**
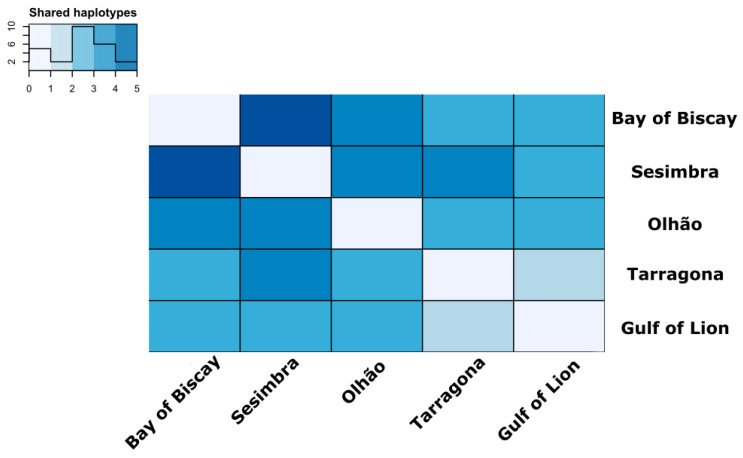
Heatmap of shared haplotypes. Quantification of the number of haplotypes shared by sampling locations. The legend refers to absolute numbers shared among locations, with the histogram line representing the frequency distribution.

**Figure 3 genes-12-00091-f003:**
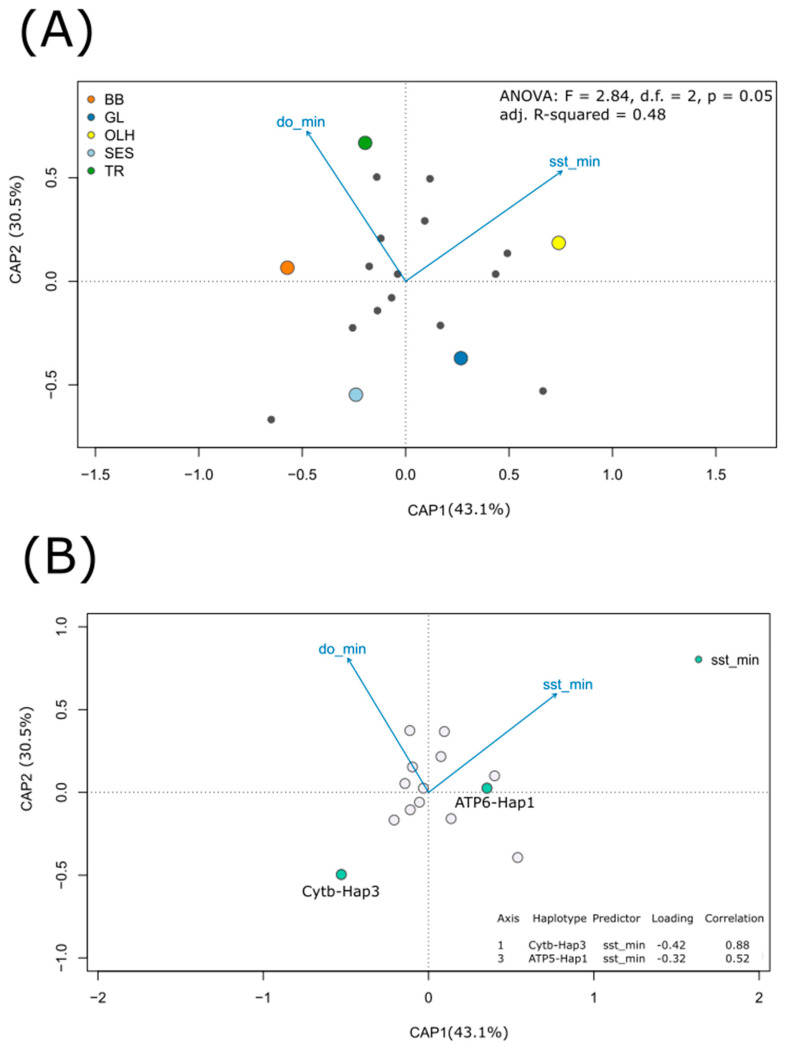
Presentation of redundancy analysis ordination with minimum values of environmental variables. (**A**) Ordination plot with haplotypes (response variables) as grey dots, populations in coloured dots and environmental variables (explanatory variables) as vectors. BB = Bay of Biscay, GL = Gulf of Lion, TR = Tarragona, OLH = Olhão, SES = Sesimbra; sst_min = minimum sea surface temperature, do_min = minimum dissolved oxygen. (**B**) Ordination plot highlighting haplotypes, Cytb-Hap3 and ATP6-Hap1, whose frequency is predicted to vary with minimum sea surface temperature (Cytb-Hap3: *r* = 0.88, *p* < 0.05; ATP6-Hap1: *r* = 0.52, *p* < 0.05), also showing respective axis and correlation coefficients.

**Table 1 genes-12-00091-t001:** Diversity estimates obtained for each mitochondrial gene region coding and respective complex of the electron transport chain.

Gene	Population	*n*	S	nHap	Hd	π	Tajima’s-D
*ATP6* Respiratory Complex V	Bay of Biscay	15	3	4	0.371	0.00135	−1.31654
	Gulf of Lion	15	0	1	0	0	na
	Olhão	18	2	2	0.11	0.00058	−1.50776
	Sesimbra	14	3	4	0.396	0.0012	−1.67053
	Tarragona	17	0	1	0	0	na
*COI* Respiratory Complex IV	Bay of Biscay	17	2	3	0.228	0.00081	−1.50358
	Gulf of Lion	20	1	2	0.1	0.00034	−1.16439
	Olhão	15	1	2	0.133	0.00046	−1.15945
	Sesimbra	14	1	2	0.143	0.00049	−1.15524
	Tarragona	18	2	3	0.216	0.00076	−1.50776
*CYTB* Respiratory Complex III	Bay of Biscay	16	3	4	0.642	0.0051	0.38767
	Gulf of Lion	17	3	4	0.676	0.00563	0.78185
	Olhão	16	4	5	0.717	0.00589	−0.05743
	Sesimbra	14	4	5	0.769	0.00716	0.476
	Tarragona	15	4	5	0.781	0.00625	0.07027
*ND5* Respiratory Complex I	Bay of Biscay	19	4	5	0.696	0.00284	−0.53717
	Gulf of Lion	20	1	2	0.521	0.00158	1.53133
	Olhão	19	2	3	0.526	0.0017	−0.04521
	Sesimbra	13	2	3	0.615	0.0021	0.2084
	Tarragona	18	4	5	0.66	0.00273	−0.67309

*n* = number of individuals per population; S = segregation sites; nHap = number of haplotypes; Hd = Haplotype diversity; π = nucleotide diversity.

**Table 2 genes-12-00091-t002:** Model results obtained with the *anova.cca* function for each pair of parameter sea surface temperature (T °C) and dissolved oxygen (mol·m^−3^). Significant models are highlighted in bold.

**Maximum Temperature and Dissolved Molecular Oxygen**				
	df	SS	F	*p* (>F)
model	2	0.366	1.858	0.117
Residuals	2	0.197	-	-
**Minimum Temperature and Dissolved Molecular Oxygen**				
	df	SS	F	*p* (>F)
model	2	0.416	**2.841**	**0.050**
Residuals	2	0.146	-	-
**Range of Temperature and Dissolved Molecular Oxygen**				
	df	SS	F	*p* (>F)
model	2	0.150	1.087	0.483
Residuals	2	0.413	-	-
**Mean of Temperature and Dissolved Molecular Oxygen**				
	df	SS	F	*p* (>F)
model	2	0.202	0.560	0.867
Residuals	2	0.361	-	-

Model formula: Haplotype frequencies ~ SST+ DO. In bold, values significant for *p* ≤ 0.05.

**Table 3 genes-12-00091-t003:** ANOVA on the minimum SST and DO model to test the marginal effect of terms with 500 permutations to access.

	**df**	**SS**	**F**	***p* (>F)**
CAP1	1	0.244	3.343	**0.050**
CAP2	1	0.171	2.339	0.150
Residual	2	0.146		
	**df**	**SS**	**F**	***p* (>F)**
sst_min	1	0.230	3.109	**0.033**
do_min	1	0.202	2.756	0.058
Residual	2	0.146		

In bold, values significant for *p* ≤ 0.05.

**Table 4 genes-12-00091-t004:** Phylogenetic inferences on episodic or pervasive historical events of selection and adopting three different algorithms.

Method	Gene	Codon	Site	*p* Value	pp [α > β]
SLAC	*CYTB*	14	42	**0.1**	n/a
		15	45	**0.1**	n/a
		16	48	**0.1**	n/a
		26	78	**0.1**	n/a
		40	120	**0.1**	n/a
		48	144	**0.1**	n/a
		53	159	**0.1**	n/a
		57	171	**0.1**	n/a
FEL	*ATP6*	53	159	**0.1**	n/a
	*ND5*	58	174	**0.1**	n/a
FUBAR	*ATP6*	24	72	n/a	**0.9**
		49	147	n/a	**0.9**
		36	108	n/a	**0.9**
		52	156	n/a	**0.9**
		53	159	n/a	**0.9**
	*ND5*	58	174	n/a	**0.9**
		64	192	n/a	**0.9**
		90	270	n/a	**0.9**
	*COI*	8	24	n/a	**0.9**
		44	132	n/a	**0.9**
		53	159	n/a	**0.9**
		57	171	n/a	**0.9**

Only sites whose deviations from the null hypothesis from neutral evolution were reported and are shown. Significance was determined with threshold for *p* ≤ 0.1, in likelihood models SLAC and FEL, and with posterior probabilities in FUBAR. Only sites denoting significant values (in bold) are reported. Note that the *p*-value on SLAC relates to P (dN/Ds < 1). All sites are reporting synonymous substitutions. n/a denotes statistic not available in the specific test, as defined in the Methods Section.

**Table 5 genes-12-00091-t005:** Diversity estimates regarding the five microsatellite markers.

Locations	*n*	rAR	pA	H_o_	H_e_
Bay of Biscay	20	13.44	2.24	0.71	0.86
Gulf of Lion	20	12.41	1.80	0.73	0.83
Olhão	20	12.89	2.10	0.74	0.86
Sesimbra	16	12.00	1.78	0.65	0.84
Tarragona	20	12.18	1.18	0.63	0.80

*n* = number of individuals; rAR = rarefied allelic richness; pA = private alleles; H_o_ = Observed heterozygosity and H_e_ = Expected heterozygosity.

## Data Availability

Sequence data is available as supplementary file. Sequences are deposited in NCBI under the accession numbersMW437948-MW438277.

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
