# Peer review of "Targeted Sequencing of Mitochondrial Genes Reveals Signatures of Molecular Adaptation in a Nearly Panmictic Small Pelagic Fish Species"

_genes, 2021, doi:10.3390/genes12010091_

Round 1
Reviewer 1 Report
The authors used seascape genetic analysis to evaluate the genetic differences/variability of the mitochondrial genome of European sardine Sardina pilchardus distributed around the Iberia Peninsula, according to spatial and environmental parameters (Oxygen and temperature).
The authors assumed that their samples originated from one population with all possible exchange of genetic materials between fish harvested from different fishing grounds. If it is assumed that this population is panmictic, then any differences in the mitochondrial genome between different fishing grounds can be attributed to environmental (spatial) and not a genetic component.
The authors observed the dominance of some haplotypes on the low range of thermal conditions, indicating a behavioural response associated with genetic features.
I can see the value of the present work and the rationale of the experimental design.
There are significant differences in the environmental conditions in the Atlantic and the Mediterranean waters, so I congratulate the authors on their experimental design.
I also agree with the genetic markers selected in this work, all are know to be relevant for aerobic metabolism and their dependence on temperature is also significant.
I however have an issue regarding the assumption of a panmictic population structure.
For example, geographic isolation can limit the exchange of genetic material between different geographic location as indeed is the case in the Atlantic and the Mediterranean waters. See for example the work of Ramon, M. M., & Castro, J. A. (1997). Genetic variation in natural stocks of Sardina pilchardus (sardines) from the western Mediterranean Sea. Heredity, 78(5), 520-528.
For this reason, I believe that this manuscript needs a revision; the authors should acknowledge the limitation of their work but also the problem of working with mitochondrial genome in terms of spatial genetic variability.
This work has valuable information, but a revision is required to properly base the conclusions on the findings.
For this reason a major revision is required, which will avoid the assumption of a panmixia of the population and focus on the spatial variability of the mitochondrial genome and how this relates to environmental conditions and the capacity for aerobic metabolism.
Author Response
The authors used seascape genetic analysis to evaluate the genetic differences/variability of the mitochondrial genome of the European sardine Sardina pilchardus distributed around the Iberia Peninsula, according to spatial and environmental parameters (oxygen and temperature). The authors assumed that their samples originated from one population with all possible exchange of genetic materials between fish harvested from different fishing grounds. If it is assumed that this population is panmictic, then any differences in the mitochondrial genome between different fishing grounds can be attributed to environmental (spatial) and not a genetic component. The authors observed the dominance of some haplotypes on the low range of thermal conditions, indicating a behavioural response associated with genetic features. I can see the value of the present work and the rationale of the experimental design. There are significant differences in the environmental conditions in the Atlantic and the Mediterranean waters, so I congratulate the authors on their experimental design. I also agree with the genetic markers selected in this work, all are known to be relevant for aerobic metabolism and their dependence on temperature is also significant.
I however have an issue regarding the assumption of a panmictic population structure. For example, geographic isolation can limit the exchange of genetic material between different geographic location as indeed is the case in the Atlantic and the Mediterranean waters. See for example the work of Ramon, M. M., & Castro, J. A. (1997). Genetic variation in natural stocks of Sardina pilchardus (sardines) from the western Mediterranean Sea. Heredity, 78(5), 520-528. For this reason, I believe that this manuscript needs a revision; the authors should acknowledge the limitation of their work but also the problem of working with mitochondrial genome in terms of spatial genetic variability. This work has valuable information, but a revision is required to properly base the conclusions on the findings. For this reason a major revision is required, which will avoid the assumption of a panmixia of the population and focus on the spatial variability of the mitochondrial genome and how this relates to environmental conditions and the capacity for aerobic metabolism.
Response 1: First we would like to thank the reviewer for the kind words on our work. Regarding the consideration of panmixia as a baseline expectation for population structure, we appreciate the reviewer´s input with the above-mentioned reference, which we now added to the manuscript. Consequently, we have added text to the discussion to accommodate the population differentiation perspective of Ramon & Castro 1997. We also discussed how our study adds-on to those already existing to demand a full genome approach to resolve the stock structure of this commercially important species:
Lines 337- “Because Mantel tests did not support a general pattern of isolation by distance, we consider having identified a signature of limited dispersal among sampled locations, possibly due to the natural barrier imposed by the Alboran Sea as suggested by Ramon & Castro (1997).”
However, other studies on sardines – cited in our manuscript - rather supported a panmictic population. Due to those apparently conflicting views, we prefer to maintain the more consensual expectation of “near panmixia” or “nearly panmictic populations”. Note that doing it so does not prevent to assess the spatial variability of mitochondrial – or any other gene – at the light of environmental factors. Such analytical methods nevertheless constitute the current basis of “outlier detection” methods, which we partially intended to replicate here by assessing two distinct types of molecular markers.

Reviewer 2 Report
Review Genes-1045925
I had the pleasure of evaluating the paper entitled “Targeted sequencing of mitochondrial genes reveals signatures of molecular adaptation in a nearly panmictic small pelagic species”, by Baltazar-Soares et al.
The paper is well structured and well written, however there are inconsistencies in the analyses and discussion of the results throughout the manuscript. In addition, some affirmations do not have analytical results to back them up. Specifically, there was an inconsistency on the p-value level to which pairwise FST test were deemed significant, as well as for the methods for the investigation of selection on genes. Also, while no Isolation by Distance (IBD) test or Mantel test was carried out to test for a putative significant relation between geographic distance and genetic structuring, the authors keep mentioning the putative presence of such a phenomenon to explain the structuring they find. Finally, while they mentioned twice in the results that the genetic diversity indices were not significantly different among populations, they infirm this statement in the discussion (line 308 309).
As such I recommend not to accept the manuscript in its present form, since it appears that the population genetics and evolutionary analyses need to be rechecked, rediscussed and completed by additional analyses which I mention in the detailed review below.
Introduction:
- Line 69: replace “it thus” by “it is thus”.
Methods:
- Line 89: specify the manufacturer of the EZ-DNA kit.
- Line 120: I am guessing that by “Only fragments with matching single and paired ends were kept”, you mean “only reads that properly paired were kept”?
- Line 120: state the program you used to filter out the non-properly paired reads, low quality alignments and low coverage sites.
- You did not check for the presence of putative null alleles with Micro-checker. It is known to be a major flaw of microsatellite markers. Could that explain the deviation from HWE and the highest FST constantly found across populations for the SAR218 locus? It is not discussed here.
- Line 128-129: you do not specify which p-value will be significant for the FST on haplotypes (which I guess is p<=0.05 looking at the results section lines 208-210).
- Line 159-160: you did not mention what the significance level would be for the different methods.
- Line 185: “was reported to be THE subject OF clinal selection”, or “was reported to be subjected to clinal selection”
Results:
- Line 196: “Position of variant sites identified by vcftools”: from what you explained in the methods section, you identified the variants with Stacks, not vcftools (which actually does not do SNP calling). You might have used vcftools on the output from Stacks, in order to obtain a filtered dataset (see my comment on the lack of specification of a software to do the filtering step in the methods).
- Line 197: on the reconstruction of haplotypes from the reference mitogenome: did you encounter individuals for which you had more than one possible haplotype? (e.g. heterozygous individuals?). I would like to know more about the specificities of your haplotypic reconstruction (probably to be placed in the material and methods).
- Line 212-213: you could test for this effect with IBD or Mantel Test.
- Lines 238-245: the end of the caption of Figure 3 does not have the proper font size.
- Lines 247: Table 2 titles need to be reformatted (all in bold or all not in bold).
- Tables in general: please choose between bold values and stars for significance. In table 4 for example, it seems that only the FEL-ATP6 value is significant (star) while you mention in the legend that significance is at p=<0.05 (meaning that all of the results presented in Table 4 are significant…).
- Lines 253-261 and Table 4: In the text, you say that the significance is at posterior probability of α>β, >= 0.9, and present a list of sites for each gene, but in the table 4 you show results that are <0.9 for some of those sites. I do not understand.
- Lines 283-284: I though you considered significance for FST values at p < 0.01 (according to your statement in the material and methods lines 180-181). As such, the FST value of 0.05 between Bay of Biscay and Gulf of Lion is not significant (p=0.02). Especially, you state the significance at p < 0.01 again lines 291-292…
Discussion:
- Lines 308-309: I don’t understand this, because in the results you kept mentioning that the genetic diversity estimates were not significantly different across populations for the microsatellites (line 268) and that only ATP6 showed population differences for the haplotypes (line 206-208). As such, you cannot say that “the data indicated a trend”, since it apparently is not statistically significant.
- Lines 319-320 and 322-323: the problem is you didn’t test for isolation by distance with your microsatellite markers. You should try to test for it since you have all the necessary data to do so.
- With the problems regarding statistical significance of some tests you carried out, I fear that most of the discussion might be irrelevant here, as long as these tests are not checked again.
Author Response
I had the pleasure of evaluating the paper entitled “Targeted sequencing of mitochondrial genes reveals signatures of molecular adaptation in a nearly panmictic small pelagic species”, by Baltazar-Soares et al. The paper is well structured and well written, however there are inconsistencies in the analyses and discussion of the results throughout the manuscript. In addition, some affirmations do not have analytical results to back them up. Specifically, there was an inconsistency on the p-value level to which pairwise FST test were deemed significant, as well as for the methods for the investigation of selection on genes.
Response 1: We would like to thank the reviewer for the feedback provided to our work, we believe that tackling their comments will greatly improve the content of our manuscript. We would also like to apologize on our inconsistency in reporting significant values. We have utilized many different statistics holding on different principles and recommended thresholds for significance and incorrectly, have attempted to be consistent across reporting significance. All issues have now been tackled.
Regarding the p-values reported for FST pairwise comparisons, the reviewer stands correct: we had discrepancy in between what was considered significant for mitochondrial and microsatellite data, though the number of pairwise comparisons remain the same. At the light of other reviewer comments – who stressed out the population differentiation component of our work – we decided that to better contextualize our findings on the existing body of literature, to main the p=0.05 as threshold for FST pairwise comparisons. Results, methods and discussion were updated accordingly.
Also, while no Isolation by Distance (IBD) test or Mantel test was carried out to test for a putative significant relation between geographic distance and genetic structuring, the authors keep mentioning the putative presence of such a phenomenon to explain the structuring they find.
Response 2: We have performed the test but because results were non-significant, we did not present it. We have now reported the values.
Finally, while they mentioned twice in the results that the genetic diversity indices were not significantly different among populations, they infirm this statement in the discussion (line 308 309).
As such I recommend not to accept the manuscript in its present form, since it appears that the population genetics and evolutionary analyses need to be rechecked, rediscussed and completed by additional analyses which I mention in the detailed review below.
Response 3: We have corrected it accordingly
Introduction:
- Line 69: replace “it thus” by “it is thus”.
Response 3: Corrected
Methods:
- Line 89: specify the manufacturer of the EZ-DNA kit.
Response 4: Corrected
- Line 120: I am guessing that by “Only fragments with matching single and paired ends were kept”, you mean “only reads that properly paired were kept”?
Response 5: Corrected
- Line 120: state the program you used to filter out the non-properly paired reads, low quality alignments and low coverage sites.
Response 6: We used BWA, as stated in lines 120-121. The filtering parameters mentioned in those lines were input on the script utilized to process and align reads against sardine´s mitochondrial genome. As such, the outfile of BWA, for each individual, consisted only of reads that met the above-mentioned criteria. Remaining filters relate to those input on Stacks default pipeline, but also with VCFtools, which we used to filter out low coverage sites (<100). This last piece of information was not present in the text
- You did not check for the presence of putative null alleles with Micro-checker. It is known to be a major flaw of microsatellite markers. Could that explain the deviation from HWE and the highest FST constantly found across populations for the SAR218 locus? It is not discussed here.
Response 7: The reviewer stands correct, and we have now performed a Micro-Checker analysis. It indeed revealed that the SAR218 is consistently reporting null alleles across all analysed populations. We have now included a table resuming Micro-Checker results in the supplementary material and changed our discussion to accommodate the SAR218 report.
- Line 128-129: you do not specify which p-value will be significant for the FST on haplotypes (which I guess is p<=0.05 looking at the results section lines 208-210).
Response 8: Corrected
- Line 159-160: you did not mention what the significance level would be for the different methods.
Response 9: We have specified the significance method for each test on the legend of table 4, but we agree to be better to have it on the methods already – it has now been copied to this section of the manuscript.
- Line 185: “was reported to be THE subject OF clinal selection”, or “was reported to be subjected to clinal selection”
Response 10: Corrected
Results:
- Line 196: “Position of variant sites identified by vcftools”: from what you explained in the methods section, you identified the variants with Stacks, not vcftools (which actually does not do SNP calling). You might have used vcftools on the output from Stacks, in order to obtain a filtered dataset (see my comment on the lack of specification of a software to do the filtering step in the methods).
Response 11: Indeed, variants were identified in Stacks though the position – in relation to the reference genome – not with vcftools; though obviously extracted from the .vcf file. We have changed it for the correct software. Indeed, we used VCFtool to filer for low coverage variants, please see our reply on the previous section.
Line 197: on the reconstruction of haplotypes from the reference mitogenome: did you encounter individuals for which you had more than one possible haplotype? (e.g. heterozygous individuals?). I would like to know more about the specificities of your haplotypic reconstruction (probably to be placed in the material and methods).
Response 12: We did not take specific measures to accurately identify heteroplasmy, i.e. selectively amplifying the mtDNA by long-range PCR, isolating the mtDNA using mtDNA enrichment kits, or isolating the mitochondria themselves prior to DNA extraction. Without large numbers of sardine´s mitochondrial genomes available (there are only two) we were not confident to perform deeper analysis on this topic. As such, and due to filtering out low coverage reads to ensure our variants were not due to sequencing errors, we not surprisingly found only 1/1 variants.
- Line 212-213: you could test for this effect with IBD or Mantel Test.
Response 12: We have now checked for IBD with Mantel tests implemented in R´s package ecodist. Methods, results and discussion were updated accordingly.
- Lines 238-245: the end of the caption of Figure 3 does not have the proper font size.
Response 13: Corrected
- Lines 247: Table 2 titles need to be reformatted (all in bold or all not in bold).
Response 14: Corrected
- Tables in general: please choose between bold values and stars for significance. In table 4 for example, it seems that only the FEL-ATP6 value is significant (star) while you mention in the legend that significance is at p=<0.05 (meaning that all of the results presented in Table 4 are significant…).
Response 15: We apologize for the inconsistency on this aspect. We kept bold for significant results in all tables. And indeed yes, Table 4. only shows the sites whose deviations from null hypothesis from neutral evolution were reported. We have now clearly stated it in text.
- Lines 253-261 and Table 4: In the text, you say that the significance is at posterior probability of α>β, >= 0.9, and present a list of sites for each gene, but in the table 4 you show results that are <0.9 for some of those sites. I do not understand.
Response 16: The values were supposed to be rounded to the decimal – original output of the software – as thus >=0.85 are reported as significant observations. We preferred to present 3 decimals for consistency, though we now realize it did not help. We decided to resort back to the original one decimal representation.
- Lines 283-284: I though you considered significance for FST values at p < 0.01 (according to your statement in the material and methods lines 180-181). As such, the FST value of 0.05 between Bay of Biscay and Gulf of Lion is not significant (p=0.02). Especially, you state the significance at p < 0.01 again lines 291-292…
Response 17: Please see our first reply on this topic.
Discussion:
- Lines 308-309: I don’t understand this, because in the results you kept mentioning that the genetic diversity estimates were not significantly different across populations for the microsatellites (line 268) and that only ATP6 showed population differences for the haplotypes (line 206-208). As such, you cannot say that “the data indicated a trend”, since it apparently is not statistically significant.
Response 18: We apologize for the misunderstanding, the error stems from our choice of words/expressions on line 268 (“significant”) and 309 (“indicated a clear trend). We replaced those with “prominent” and “suggested”, respectively. Data indeed showed that sardines caught on the Bay of Biscay consistently reported the highest levels of genetic diversity on analysed parameters, and that was the point we intended to make on this paragraph.
- Lines 319-320 and 322-323: the problem is you didn’t test for isolation by distance with your microsatellite markers. You should try to test for it since you have all the necessary data to do so.
Response 19: Isolation by distance has now been performed, and we have updated the discussion in line with obtained results.
- With the problems regarding statistical significance of some tests you carried out, I fear that most of the discussion might be irrelevant here, as long as these tests are not checked again.
Response 20: We hope all concerns regarding statistical analysis have now been tackled.

Round 2
Reviewer 1 Report
I am convinved by the authors that the evidence presented here at least partially support a near panmixia of the fish population studied here. I am also satisfied with the changes and corrections in the revised version of this manuscript. The data are crucial to study and manage the stock of this species in the region.
Author Response
We would like once again to thank the reviewer for the feedback and suggestions that very much helped to improve our manuscript.